# Model Zoos: A Dataset of Diverse Populations of Neural Network Models

**Konstantin Schürholt**[1]**, Diyar Taskiran**[1]**, Boris Knyazev**[2]**, Xavier Giró-i-Nieto**[3]**, Damian Borth**[1]

[1] AIML Lab, School of Computer Science, University of St.Gallen
[2] Samsung - SAIT AI Lab, Montreal
[3] Institut de Robòtica i Informàtica Industrial, Universitat Politècnica de Catalunya

## Abstract

In the last years, neural networks (NN) have evolved from laboratory environments to the state-of-the-art for many real-world problems. It was shown that NN models (i.e., their weights and biases) evolve on unique trajectories in weight space during training. Following, a population of such neural network models (referred to as *model zoo*) would form structures in weight space. We think that the geometry, curvature and smoothness of these structures contain information about the state of training and can reveal latent properties of individual models. With such *model zoos*, one could investigate novel approaches for (i) model analysis, (ii) discover unknown learning dynamics, (iii) learn rich representations of such populations, or (iv) exploit the *model zoos* for generative modelling of NN weights and biases. Unfortunately, the lack of standardized *model zoos* and available benchmarks significantly increases the friction for further research about populations of NNs. With this work, we publish a novel dataset of *model zoos* containing systematically generated and diverse populations of NN models for further research. In total the proposed model zoo dataset is based on eight image datasets, consists of 27 *model zoos* trained with varying hyperparameter combinations and includes 50'360 unique NN models as well as their sparsified twins, resulting in over 3'844'360 collected model states. Additionally, to the model zoo data we provide an in-depth analysis of the zoos and provide benchmarks for multiple downstream tasks. The dataset can be found at www.modelzoos.cc.

## 1 Introduction

The success of Neural Networks (NN) is surprising, considering the hard optimization problem to be solved during training of NNs. Specifically, NN training is NP-complete [2], the loss surface and optimization problem are non-convex [9, 17, 31] and the parameter space to fit during training is high dimensional [3]. Additionally, NN training is sensitive to random initialization and hyperparameter selection [19, 32]. Together, this leads to an interesting characteristic of NN training: given a dataset and an architecture, different random initializations or hyperparameters lead to different minima on the loss surface and therefore result in different model parameters (i.e., weights and biases). Consequently, multiple training results in different NN models. The resulting population of NN (referred to as *model zoo*) is an interesting object to study: Do individual models of the model zoo have something in common? Do they form structures in weight space? What can we infer from such structures? Can we learn representations of them? Lastly, can such structures be exploited to generate new models with controllable properties?

These questions have been partially answered in prior work. Theoretical and empirical work demonstrates increasingly well-behaved loss surfaces for growing number of parameters [17, 8, 32]. The shape of the loss surface and the starting point is determined by hyperparameters and the initialization,

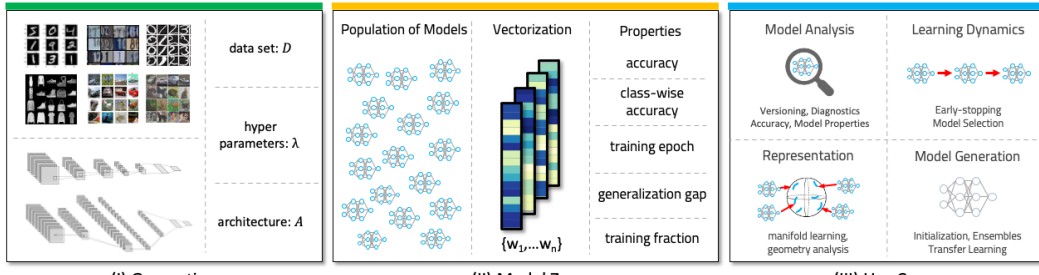

Figure 1: The proposed dataset of model zoos is trained on several image dataset with two CNN architectures and a multiple configurations of hyperparameters. The resulting population of neural network models is vectorized and made available with all meta-data such as the generating factors of the model zoo as well as the model properties such as accuracy, generalization gap and other. Potential use cases are (a) model property prediction, (b) inference of learning dynamics, (c) representation learning, or (d) model generation.

respectively [32]. NN training navigates the loss surface with iterative, gradient-based update schemes smoothed by momentum. The step length along a trajectory as well as the curvature are determined by the change of the loss as well as how aligned the subsequent updates are [4, 49]. Together, these findings suggest that populations of NN models evolve on unique and smooth trajectories in weight space. Related work has empirically confirmed the existence of such structures in NNs [11], demonstrated the feasibility to learn representations of them, showed that they encode information on model properties [55, 12, 50] and can be used to generate unseen models with desirable properties [52, 51, 64, 26] To thoroughly answer the questions above, a large and systematically created dataset of model weights is necessary.

Unfortunately, so far only few model zoos with specific properties have been published [55, 12, 54, 50]. While many machine learning domains have standardized datasets, there is no model zoo nor a benchmark to evaluate and compare against. The lack of a standardized model zoos has three significant disadvantages: (i), existing model zoos are usually designed for a specific purpose and of limited general utility. Their design space is rather sparse, covering only small portions of all available hyperparameter combinations. Moreover, some existing zoos are generated on synthetic tasks and are small, containing only a small population of models; (ii), researchers have to choose between using an existing zoo or generating a new one for each new experiment, weighing disadvantages of existing zoos against the effort and computational resources required to generate a new zoo; (iii), a new model zoo causes subsequent work to lose comparability to existing research. Therefore, the lack of a benchmark model zoo significantly increases the friction for new research.

**Our contributions:** To study the behaviour of populations of NNs, we publish a large-scale model zoo of diverse populations of neural network models with controlled generating factors of model training. Special care has been taken in their design and the used protocols for training. To do so, we have defined and restricted the generating factors of model zoo training to achieve desired zoo characteristics.

The zoos are trained on eight standard image classification datasets, with a broad range of hyperparameters and contain thousands of configurations. Further, we add sparsified *model zoo twins* to each of these zoos. All together, the zoos include a total of 50'360 unique image classification NNs, resulting in over 3'844'360 collected model states.

Potential use-cases for the model zoo include (a) model analysis for reliability, bias, fairness, or adversarial vulnerability, (b) inference of learning dynamics for efficiency gain, model selection or early stopping, (c) representation learning of such populations, or (d) model generation. Additionally, we present an analysis of the model zoos and a set of experimental setups for benchmarks on these use-cases and initial results as foundation for evaluation and comparison.

With this work we provide a standardized dataset of diverse model zoos connected to popular image datasets, its corresponding meta-data and performance evaluations to the machine learning research community. All data is made publicly available to foster community building around the topic and to provide a ground for use beyond the defined benchmark tasks. An overview of the proposed dataset and benchmark as well as potential use-cases can be found in Fig. 1.

## 2 Existing Populations of Neural Networks Models

With the increase in usage of neural networks, requirements for evaluation, testing and certification have grown. Methods to analyze NN models may attempt to visualize salient features for a given class [62, 25, 61], investigate the robustness of models to specific types of noise [66, 7], predict model properties from model features [58, 24, 6] or compare models based on their activations [46, 42, 44] However, while most of these methods rely on common (image) datasets to train and evaluate their models, there is no common dataset of neural network models to compare the evaluation methods on. Model zoos as common evaluation datasets can be a step up to evaluate the evaluation methods.

There are only few publications who use model zoos. In [35], zoos of pre-trained models are used as teacher models to train a target model. Similarly, [53] propose a method to learn a combination of the weights of models from a zoo for a new task. [65] uses a zoo of GAN models trained with different methods to accelerate GAN training. To facilitate continual learning, [47] propose to generate zoos of models trained on different tasks or experiences, and to ensemble them for future tasks.

Larger model zoos containing a few thousand models are used in [55] to predict the accuracy of the models from their weights. Similarly, [12] use zoos of larger models to predict hyperparameters from the weights. In [16], a large collection of 3x3 convolutional filters trained on different datasets is presented and analysed. Other work identifies structures in the form of subspaces with beneficial properties [37, 56, 1]. [50] use zoos to learn self-supervised representations on the weights of the models in the zoo. The authors demonstrate that the learned representations have high predictive capabilities for model properties such as accuracy, generalization gap, epoch and various hyperparameters. Further, they investigate the impact of the generating factors of model zoos on their properties. [52, 51] demonstrate that learned representations can be instantiated in new models, as initialization for fine-tuning or transfer learning. This work systematically extends their zoos to more datasets and architectures.

## 3 Model Zoo Generation

The proposed model zoo datasets contain systematically generated and diverse populations of neural networks. Since the applicability of the model zoos for downstream tasks largely depends on the composition and properties of the zoos, special care has to be taken in their design and the used protocol for training. The entire procedure can be considered as defining and restricting the generating factors of model zoo training with respect to their latent relation of desired zoo characteristics. The described procedure and protocol could be also used as general blueprint for the generation of model zoos.

In our paper, the term architecture means the structure of a NN, i.e., set of operations and their connectivity. We use 'model' to denote an instantiating of an architecture with weights over all stages of training, 'model state' to denote the model with the specific state of weights at a specific training epoch, and the weights $\mathbf{w}$ to denote all trainable parameters (weights and biases).

### 3.1 Model Zoo Design

**Generating Factors**   Following [55], we define the tuple $\{\mathcal{D}, \lambda, \mathcal{A}\}$ as a configuration of a model zoo's generating factors. We denote the dataset of image samples with their corresponding labels as $\mathcal{D}$. The NN architecture is denoted by $\mathcal{A}$. We denote the set of hyperparameters used for training, (*e.g.*, loss function, optimizer, learning rate, weight initialization, seed, batch-size, epochs) as $\lambda$. While dataset $\mathcal{D}$ and architecture $\mathcal{A}$ are fixed for a model zoo, $\lambda$ provides not only the set of hyperparameters but also configures the ranges for individual hyperparameter such as learning rate for model zoo generation. Training with such differing configurations $\{\mathcal{D}, \lambda, \mathcal{A}\}$ results in a population of NN models i.e., the model zoo. We convert the weights and biases of each model to a vectorized form. In the resulting model zoo $\mathcal{W} = \{\mathbf{w}_1, ...., \mathbf{w}_M\}$, $\mathbf{w}_i$ denotes the flattened vector of the weights and biases of one trained NN model from the set of $M$ models of the zoo.

**Configurations & Diversity**   The model zoos have to be representative of real world models, but also diverse and span an interesting range of properties. The definition of diversity of model zoos, as well as the choice of how much diversity to include, is as difficult as in image datasets, e.g. [10, 13]. Model zoos can be diverse in their properties (i.e., performance) as well as in their generating factors $\lambda$, or in their weights $\mathbf{w}$. We aim at generating model zoos with a rich set of models and diversity in these aspects. As these zoo properties are effects of the generating factors, we tune the diversity of the generating factors and evaluate the diversity in Section 4.

Table 1: Generating factors of the model zoos. Several values for each parameter define the grid. `Arch` denotes the architecture: CNN (s) - small CNN architecture, CNN (m) - medium CNN architecture, RN-18 - ResNet-18. `Init` denotes the initalization methods: U - uniform, N - normal, KU - Kaiming Uniform, KN - Kaiming Normal. `Activation` denotes the activation function: T - Tanh, S - Sigmoid, R - ReLU, G - GeLU. `Optim` denotes the optimizer: AD - Adam, SGD - Stochastic Gradient Descent. Models with learning rates denoted with * have been trained with a one-cycle LR scheduler, the listed LR is the maximum value.

| Dataset | Arch | Config | Init | Activation | Otpim | LR | WD | Dropout | Seed |
|---|---|---|---|---|---|---|---|---|---|
| | CNN (s) | Seed | U | T | AD | 3e-4 | 0 | 0 | 1-1000 |
| MNIST | CNN (s) | Hyp-10-r | U, N, KU, KN | T, S, R, G | AD, SGD | 1e-3,1e-4 | 1e-3, 1e-4 | 0, 0.5 | $\sim 10$ |
| | CNN (s) | Hyp-10-f | U, N, KU, KN | T, S, R, G | AD, SGD | 1e-3,1e-4 | 1e-3, 1e-4 | 0, 0.5 | 1-10 |
| | CNN (s) | Seed | U | T | AD | 3e-4 | 0 | 0 | 1-1000 |
| F-MNIST | CNN (s) | Hyp-10-r | U, N, KU, KN | T, S, R, G | AD, SGD | 1e-3,1e-4 | 1e-3, 1e-4 | 0, 0.5 | $\sim 10$ |
| | CNN (s) | Hyp-10-f | U, N, KU, KN | T, S, R, G | AD, SGD | 1e-3,1e-4 | 1e-3, 1e-4 | 0, 0.5 | 1-10 |
| | CNN (s) | Seed | U | T | AD | 3e-3 | 0 | 0 | 1-1000 |
| SVHN | CNN (s) | Hyp-10-r | U, N, KU, KN | T, S, R, G | AD, SGD | 1e-3,1e-4 | 1e-3, 1e-4, 0 | 0, 0.3, 0.5 | $\sim 10$ |
| | CNN (s) | Hyp-10-f | U, N, KU, KN | T, S, R, G | AD, SGD | 1e-3,1e-4 | 1e-3, 1e-4, 0 | 0, 0.3, 0.5 | 1-10 |
| | CNN (s) | Seed | U | T | AD | 3e-4 | 1e-3 | 0 | 1-1000 |
| USPS | CNN (s) | Hyp-10-r | U, N, KU, KN | T, S, R, G | AD, SGD | 1e-3,1e-4 | 1e-2, 1e-3 | 0, 0.5 | $\sim 10$ |
| | CNN (s) | Hyp-10-f | U, N, KU, KN | T, S, R, G | AD, SGD | 1e-3,1e-4 | 1e-2, 1e-3 | 0, 0.5 | 1-10 |
| | CNN (s) | Seed | KU | G | AD | 1e-4 | 1e-2 | 0 | 1-1000 |
| CIFAR10 | CNN (s) | Hyp-10-r | U, N, KU, KN | T, S, R, G | AD, SGD | 1e-3 | 1e-2, 1e-3 | 0, 0.5 | $\sim 10$ |
| | CNN (s) | Hyp-10-f | U, N, KU, KN | T, S, R, G | AD, SGD | 1e-3 | 1e-2, 1e-3 | 0, 0.5 | 1-10 |
| | CNN (m) | Seed | KU | G | AD | 1e-4 | 1e-2 | 0 | 1-1000 |
| CIFAR10 | CNN (m) | Hyp-10-r | U, N, KU, KN | T, S, R, G | AD, SGD | 1e-3 | 1e-2, 1e-3 | 0, 0.5 | $\sim 10$ |
| | CNN (m) | Hyp-10-f | U, N, KU, KN | T, S, R, G | AD, SGD | 1e-3 | 1e-2, 1e-3 | 0, 0.5 | 1-10 |
| | CNN (s) | Seed | KU | T | AD | 1e-4 | 1e-3 | 0 | 1-1000 |
| STL (s) | CNN (s) | Hyp-10-r | U, N, KU, KN | T, S, R, G | AD, SGD | 1e-3,1e-4 | 1e-2, 1e-3 | 0, 0.5 | $\sim 10$ |
| | CNN (s) | Hyp-10-f | U, N, KU, KN | T, S, R, G | AD, SGD | 1e-3,1e-4 | 1e-2, 1e-3 | 0, 0.5 | 1-10 |
| | CNN (m) | Seed | KU | T | AD | 1e-4 | 1e-3 | 0 | 1-1000 |
| STL | CNN (m) | Hyp-10-r | U, N, KU, KN | T, S, R, G | AD, SGD | 1e-3,1e-4 | 1e-2, 1e-3 | 0, 0.5 | $\sim 10$ |
| | CNN (m) | Hyp-10-f | U, N, KU, KN | T, S, R, G | AD, SGD | 1e-3,1e-4 | 1e-2, 1e-3 | 0, 0.5 | 1-10 |
| CIFAR10 | RN-18 | Seed | KU | R | SGD | 1e-4* | 5e-4 | 0 | 1-1000 |
| CIFAR10 | RN-18 | Seed | KU | R | SGD | 1e-4* | 5e-4 | 0 | 1-1000 |
| CIFAR10 | RN-18 | Seed | KU | R | SGD | 1e-4* | 5e-4 | 0 | 1-1000 |

Prior work discusses the impact of random seeds on properties of model zoos. While [58] use multiple random seeds for the same hyperparameter configuration, [55] explicitly argues against that to prevent information leakage between models from train to test set. To achieve diverse model zoos and disentangle the generating factors (seeds and hyperparameters), we train model zoos in three different configurations, some with random seeds, others with fixed seeds.

`Random Seeds` The first configuration, denoted as `Hyp-10-rand`, varies a broad range of hyperparameters to define a grid of hyperparameters. To include the effect of different random initializations, each of the hyperparameter nodes in the grid is repeated with ten randomly drawn seeds. One model is configured with the combination of hyperparameters and seed, with a total of ten models per hyperparameter node. It is very unlikely for two models in the zoo share the same random seed. With this, we achieve the highest amount of diversity in properties, generating factors and weights.

`Fixed Seeds` The second configuration, denoted as `Hyp-10-fix`, uses the same hyperparameter grid as , but repeats each node with ten fixed seeds [1, 2, ..., 10]. Fixing the seeds allows evaluation methods to control for the seed, isolate the influence of hyperparameter choices and still get robust results over 10 repetitions. A side effect of the (desired) isolation of factors of influence is, that fixing the seeds leads to repetitions of the starting point in weight space for models with the same seed and initialization methods. In the beginning of the training, these models may have similar trajectories.

`Fixed Hyperparameters` For the third configuration, denoted as `Seed`, we fix one set of hyperparameters, and repeat that with 1000 different seeds. With that, we achieve zoos that are very diverse in weights and covers a broad range in weight space. These zoos and can be used to evaluate the impact of weights and their starting point on model performance. The hyperparameters for the `Seed` zoos are chosen such that there is still a level of diversity in model performance.

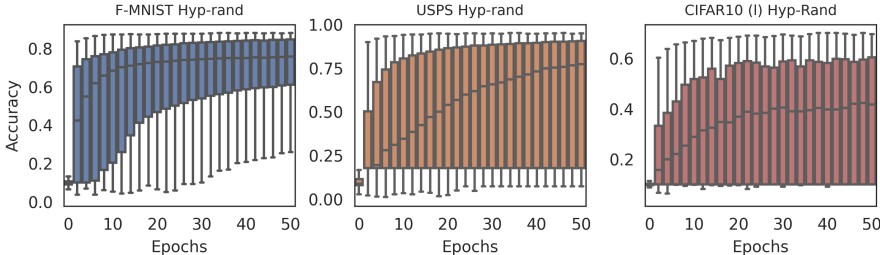

Figure 2: Accuracy distribution over epochs for the `F-MNIST Hyp-rand`, `USPS Hyp-rand` and `CIFAR Hyp-rand` zoos. All zoos show training progress and considerable performance diversity.

## 3.2 Specification of Generating Factors for Model Zoos

This section describes the systematic specification of the trained model zoos. Multiple generating factors define a configuration $\{\mathcal{D}, \lambda, \mathcal{A}\}$ for the model zoo generation, detailed in Table 1.

**Datasets** $\mathcal{D}$**:** We generate model zoos for the following image classification datasets: `MNIST` [30], `Fashion-MNIST` [57], `SVHN` [43], `CIFAR-10` [28], `STL-10` [5], `USPS` [22], `CIFAR-100` [28] and `Tiny Imagenet` [29].

**Hyperparameter** $\lambda$**:** varied hyperparameters to train models in zoos are: (1) `seed`, (2) `initialization method`, (3) `activation function`, (4) `dropout`, (4) `optimization algorithm`, (5) `learning rate`, and (6) `weight decay`. The batch-size and number of training epoch is kept constant within zoos.

**Architecture** $\mathcal{A}$**:** To preserve the comparability within a model zoo, each zoo is generated using a single neural network architecture. One of three standard architectures is used to generate each zoo. Our intention with this dataset is similar to research communities such as Neural Architecture Search (NAS), Meta-Learning or Continual Learning (CL), where initial work started small-scale [64, 47]. Hence, the first two architectures are a small and a slightly larger Convolutional Neural Network (CNN), both have three convolutional and two fully-connected layers, but different numbers of channels (details in Appendix A). The third architecture is a standard ResNet-18 [20]. The (1) `small` CNN has a total of $2'464$-$2'864$ parameters, the (2) `medium` CNN has $10'853$ parameters, the (3) ResNet-18 has 11.2M-11.3M parameters.

Compared to (1), the medium architecture (2) provides additional diversity to the collection of model zoos and performs significantly better on more complex datasets `CIFAR-10` and `STL-10`. These architectures are similar to the one used in [51]. The ResNet-18 architecture is included to apply the model zoo blueprint to models of the widely used ResNet family and so facilitate research on populations of real-world sized models.

## 3.3 Training of Model Zoos

Neural network models are trained from the previously defined three configurations $\{\mathcal{D}, \lambda, \mathcal{A}\}$ (Seed, Hyp-10-rand, Hyp-10-fix, see Sec 2.1). With the *8 image datasets* and the three configurations, this results in *27 model zoos*. The zoos include a total of around *50'360* unique neural network models.

**Training Protocol:** Every model in the collection of zoos is trained according to the same protocol. We keep the same train, validation and test splits for each zoo, and train each model for 50 epochs with gradient descent methods (SGD+momentum or ADAM). At every epoch, the model checkpoint as well as accuracy and loss of all splits are recorded. Validation and test performance are also recorded before the first training epoch. This makes 51 checkpoints per model training trajectory including the starting checkpoint representing the model initialization before training starts. The ResNet-18 zoos on CIFAR100 and Tiny Imagenet require more updates and are trained for 60 epochs. In total, this results in a set of *2'585'360* collected model states.

**Splits:** To enable comparability, this set of models is split into `training` (70%), `validation` (15%), and `test` (15%) subsets. This split is done such that all individual checkpoints of one model training (i.e., the 51 checkpoints per training) is entirely in either `training`, `validation`, or `test` and therefore no information is leaked between these subsets.

Table 2: Analysis of the diversity of our 27 model zoos (one row per zoo). Mean (std) values in % per zoo, computed on the last epoch. Agreement is computed using samples from the test split of the image dataset pairwise over the entire zoo. Higher agreement values indicate more uniform behavior and less behavioral diversity. Distance in weight space are computed pairwise over the entire zoo. Higher distance values indicate larger diversity in weight space.

| Dataset | Architecture | Config | Performance | Agreement | | Weights | | |
| | | | Accuracy | $\kappa_{aggr}$ | $\kappa_{cka}$ | $\mathbf{w}$ | l2-dist | cos dist |
|---|---|---|---|---|---|---|---|---|
| MNIST | CNN (s) | Seed | 91.1 (0.9) | 88.5 (1.3) | 77.2 (5.2) | 18.9 (58.4) | 124.1 (4.9) | 77.1 (4.1) |
| | CNN (s) | Hyp-10-r | 79.9 (30.7) | 67.7 (35.5) | 58.6 (25.9) | 0.4 (46.5) | 150.6 (66.5) | 98.8 (7.2) |
| | CNN (s) | Hyp-10-f | 80.3 (30.3) | 68.3 (35.3) | 58.8 (25.7) | 0.3 (46.7) | 149.7 (66.8) | 97.7 (10.0) |
| F-MNIST | CNN (s) | Seed | 72.7 (1.0) | 79.8 (2.6) | 82.3 (12.6) | 22.6 (55.6) | 122.0 (4.9) | 74.5 (4.4) |
| | CNN (s) | Hyp-10-r | 68.4 (23.7) | 59.9 (29.1) | 64.6 (23.5) | 1.0 (46.0) | 149.6 (62.2) | 99.2 (6.8) |
| | CNN (s) | Hyp-10-f | 68.7 (23.4) | 60.4 (28.7) | 64.6 (22.7) | 0.9 (46.3) | 148.5 (61.9) | 97.9 (9.9) |
| SVHN | CNN (s) | Seed | 71.1 (8.0) | 67.2 (10.3) | 67.7 (15.7) | 7.1 (113.7) | 137.6 (8.3) | 94.5 (5.1) |
| | CNN (s) | Hyp-10-r | 35.9 (24.3) | 61.6 (35.9) | 17.8 (28.0) | 1.4 (42.2) | 170.5 (149.4) | 83.6 (30.4) |
| | CNN (s) | Hyp-10-f | 36.0 (24.4) | 61.4 (36.0) | 18.1 (27.9) | 1.3 (42.2) | 170.0 (149.0) | 83.2 (30.7) |
| USPS | CNN (s) | Seed | 87.0 (1.7) | 87.3 (2.2) | 86.7 (6.3) | 8.2 (26.9) | 123.1 (5.2) | 75.9 (5.0) |
| | CNN (s) | Hyp-10-r | 64.7 (30.8) | 55.3 (31.4) | 50.9 (30.5) | 2.1 (39.6) | 155.5 (92.6) | 99.1 (8.9) |
| | CNN (s) | Hyp-10-f | 65.0 (30.7) | 55.4 (31.3) | 50.4 (30.4) | 1.9 (40.1) | 154.2 (93.1) | 97.3 (13.7) |
| CIFAR10 | CNN (s) | Seed | 48.7 (1.4) | 65.7 (3.1) | 72.9 (11.3) | 1.1 (11.0) | 138.7 (5.6) | 96.3 (5.1) |
| | CNN (s) | Hyp-10-r | 35.1 (16.3) | 33.3 (22.9) | 47.5 (34.0) | -0.2 (17.0) | 155.6 (71.0) | 97.5 (10.8) |
| | CNN (s) | Hyp-10-f | 35.1 (16.2) | 33.3 (22.8) | 47.3 (34.2) | -0.2 (16.9) | 155.3 (70.0) | 97.2 (11.1) |
| CIFAR10 | CNN (m) | Seed | 61.5 (0.7) | 76.0 (1.6) | 92.4 (1.7) | 0.1 (18.2) | 137.0 (7.9) | 94.1 (9.2) |
| | CNN (m) | Hyp-10-r | 39.6 (21.8) | 34.5 (27.1) | 43.2 (36.5) | -0.4 (23.0) | 158.9 (79.9) | 98.6 (12.2) |
| | CNN (m) | Hyp-10-f | 39.6 (21.7) | 34.4 (26.7) | 42.8 (37.8) | -0.4 (22.9) | 158.1 (77.2) | 98.0 (13.1) |
| STL | CNN (s) | Seed | 39.0 (1.0) | 48.4 (3.0) | 81.5 (3.9) | -0.1 (19.1) | 141.2 (5.0) | 99.8 (4.2) |
| | CNN (s) | Hyp-10-r | 23.1 (12.3) | 23.4 (20.9) | 39.0 (30.7) | 3.0 (40.0) | 158.7 (107.3) | 98.7 (10.9) |
| | CNN (s) | Hyp-10-f | 23.0 (12.2) | 23.3 (21.1) | 38.1 (30.0) | 3.0 (39.8) | 157.1 (107.2) | 96.8 (16.3) |
| STL | CNN (m) | Seed | 47.4 (0.9) | 53.9 (2.2) | 83.3 (2.3) | 0.1 (26.6) | 141.3 (6.0) | 99.9 (5.8) |
| | CNN (m) | Hyp-10-r | 24.3 (14.7) | 23.2 (24.2) | 34.1 (30.0) | 2.3 (45.7) | 159.3 (103.0) | 99.1 (12.5) |
| | CNN (m) | Hyp-10-f | 24.4 (14.7) | 23.7 (24.5) | 34.6 (30.3) | 2.3 (46.5) | 157.4 (104.1) | 97.6 (20.1) |
| CIFAR10 | ResNet-18 | Seed | 92.1 (0.2) | 93.4 (0.7) | –.- (-.-) | -0.01 (1.7) | 122.1 (3.9) | 72.2 (2.3) |
| CIFAR100 | ResNet-18 | Seed | 74.2 (0.3) | 77.6 (1.2) | –.- (-.-) | -0.1 (1.6) | 130.8 (4.1) | 83.1 (2.6) |
| Tiny ImageNet | ResNet-18 | Seed | 63.9 (0.7) | 66.1 (1.9) | –.- (-.-) | -0.1 (1.9) | 125.4 (4.9) | 77.1 (3.0) |

**Sparsified Model Zoo Twins:** Model sparsification is an effective method to reduce computational cost of models. However, methods to sparsify models to a high degree while preserving the performance are still actively researched [21]. In order to allow systematic studies of sparsification, we are extending the model zoos with sparsified *model zoo twins* serving as counterparts to existing zoos in the dataset. Using Variational Dropout (VD) [41], we sparsify the trained models from existing model zoos. VD generates a sparsification trajectory for each model, along which we track the performance, degree of sparsity and the sparsified checkpoint. With 25 sparsification epochs, this yields 1'259'000 sparsification model states.

## 3.4   Data Management and Accessibility of Model Zoos

The model zoos are made publicly available in an accessible, standardized, and well documented way to the research community under the Creative Commons Attribution 4.0 license (CC-BY 4.0). We ensure the technical accessibility of the data by hosting it on Zenodo, where the data will be hosted for at least 20 years. Further, we take steps to reduce access barriers by providing code for data loading and preprocessing, to reduce the friction associated with analyzing of the raw zoo files. All code can be found on the model zoo website www.modelzoos.cc. To ensure conceptional accessibility, we include detailed insights, visualizations and the analysis of the model zoo (Sec. 4) with each zoo. Further details can be found in Appendix B.

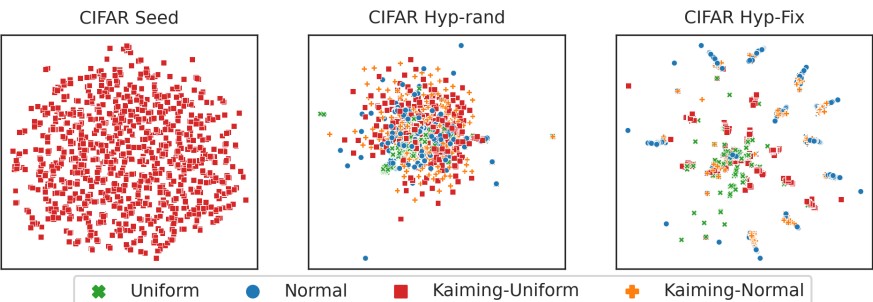

Figure 3: Visualization of the weights of the large CIFAR model zoos in different configurations. The weights are reduced to 2d using UMAP, preserving both local and global structure. In the `Seed` configuration, the UMAP reduction contains little structure. The `Hyp-rand` is equally little structured. In contrast, `Hyp-fix` contains visible clusters of initialization methods.

## 4 Model Zoo Analysis

The model zoos have been created aiming at diversity in generating factors, weights and performance. In this section, we analyse the zoos and their properties. Zoo cards with key values and visualizations are provided along with the zoos online. We consider models at their last epoch for the analysis. For all later analysis, non-viable checkpoints are excluded from each zoo. This includes the removal of every checkpoint with NaN values or values beyond a threshold. The threshold value is set for each zoo, such that it only excludes diverging models.

**Performance** To investigate the performance diversity, we consider the accuracy of the models in the zoo, see Table 2 and Figure 2. As expected, the zoos with variation only in the seed show the smallest variation in performance. Changing the hyperparameters induces a broader range of variation. Changing (Hyper-10-rand) or fixing (Hyper-10-fix) the seeds does not affect the accuracy distribution.

**Model Agreement** To get a more in-depth insights in the diversity of model behavior, we investigate their pairwise agreement, see Table 2. To that end, we compute the rate of agreement of class prediction between two models as $\kappa_{aggr} = \frac{1}{N} \sum_{1=1}^{N} \delta_{y_i}$. Here $y_i^k, y_i^l$ are the predictions of models $k, l$ for sample $i$ of $N$ samples. Further, $\delta_{y_i} = 1$ if $y_i^k = y_i^l$ and otherwise $\delta_{y_i} = 0$. Further, we compute the pairwise centered kernel alignment (cka) score between intermediate and last layer outputs and denote it as $\kappa_{cka}$. The cka score evaluates the correlation of activations, compensating for equivariances typical for neural networks [44]. In empirical evaluations, we found the cka score robust for relatively small number of image samples, and compute the scrore using 50 images to reduce the computational load. Both agreement metrics confirm the expectation and performance results. Zoos with higher overall performance naturally have a higher agreement on average, as there fewer samples on which to disagree. Zoos with varying hyperparameters(Hyp-10-rand and Hyp-10-fix) agree less on average than zoos with changes in seed only (Seed). What is more, the distribution of $\kappa_{aggr}$ and $\kappa_{cka}$ in the `Seed` zoos is unimodal and approximately gaussian. In the `Hyp-10` zoos, the distributions are bi-modal, with one mode around 0.1 (0.0) and the other around 0.9 (0.75) in hard agreement (cka score). In these zoos, models agree to a rather high degree with some models, and disagree with others.

**Weights** Lastly, we investigate the diversity of the model zoos in weight space, see again Table 2. By design, the mean weight value of the zoos varying only in the seed is larger than in the other zoos, while the standard deviation does not differ greatly (Table 2, column w). To get a better intuition in the distribution of models in weight space, we compute the pairwise $\ell_2(\mathbf{w_k}, \mathbf{w_l}) = \frac{\|\mathbf{w_k} - \mathbf{w_l}\|_2^2}{1/N \sum_{n=1}^{N} \|\mathbf{w_n}\|_2^2}$ and cosine distance $cos(\mathbf{w_k}, \mathbf{w_l}) = 1 - \frac{\mathbf{w_l}^T \mathbf{w_k}}{\|\mathbf{w_k}\|_2^2 \|\mathbf{w_l}\|_2^2}$, and investigate their distribution. Here, too, varying the hyperparameters introduces higher amounts of diversity, while changing or fixing the seeds does not affect the weight diversity much. As these values are computed at the end of model training, repeated starting points due to fixed seeds appear not to reduce weight diversity significantly. In a more hands-off approach, we compute 2d reductions of the weight over all epochs using UMAP [40]. In the 2d reductions (see Figure 3), the zoos varying in seed only show little to no structure.

Table 3: Benchmark results for predicting model properties from the weights ($\mathbf{w}$) and layer-wise weight statistics ($s(\mathbf{w})$) using linear models. We report the prediction $R^2$ for accuracy, generalization gap (GGap), epoch, learning rate (LR) and dropout (Drop) and prediction accuracy for initalization method (Init) and activation function (Act). Values reported in %, higher values are better.

| Dataset | Architecture | Config | Accuracy | | GGap | | Epoch | | Init | | Act | |
|---|---|---|---|---|---|---|---|---|---|---|---|---|
| | | | $\mathbf{w}$ | $s(\mathbf{w})$ | $\mathbf{w}$ | $s(\mathbf{w})$ | $\mathbf{w}$ | $s(\mathbf{w})$ | $\mathbf{w}$ | $s(\mathbf{w})$ | $\mathbf{w}$ | $s(\mathbf{w})$ |
| MNIST | CNN (s) | Seed | 92.3 | 98.7 | 2.1 | 68.8 | 87.2 | 97.8 | n/a | n/a | n/a | n/a |
| | CNN (s) | Hyp-10-r | -11.2 | 69.4 | -49.8 | 13.7 | -95.5 | 14.3 | 42.6 | 77.6 | 45.5 | 78.5 |
| | CNN (s) | Hyp-10-f | 66.5 | 70.1 | 5.4 | 12.5 | -4.8 | 14.5 | 94.3 | 79.8 | 81.2 | 76.8 |
| F-MNIST | CNN (s) | Seed | 87.5 | 97.2 | 20.9 | 60.5 | 89.1 | 97.1 | n/a | n/a | n/a | n/a |
| | CNN (s) | Hyp-10-r | 8.7 | 76.9 | -47.5 | 13.7 | -70.1 | 18.9 | 48.4 | 81.5 | 47.9 | 79.6 |
| | CNN (s) | Hyp-10-f | 62.4 | 75.6 | 3.9 | 12.6 | -2.0 | 17.0 | 95.4 | 81.6 | 84.6 | 77.7 |
| SVHN | CNN (s) | Seed | 91.0 | 98.6 | -42.8 | 65.9 | 66.9 | 92.5 | n/a | n/a | n/a | n/a |
| | CNN (s) | Hyp-10-r | -8.6 | 90.3 | -55.3 | 27.6 | -30.5 | 11.1 | 38.2 | 58.5 | 55.7 | 72.3 |
| | CNN (s) | Hyp-10-f | 64.2 | 89.9 | 17.5 | 27.4 | -0.1 | 11.1 | 67.3 | 58.2 | 76.1 | 73.6 |
| USPS | CNN (s) | Seed | 92.5 | 98.7 | 44.3 | 71.8 | 86.0 | 98.4 | n/a | n/a | n/a | n/a |
| | CNN (s) | Hyp-10-r | -11.5 | 70.3 | -35.2 | 13.6 | -75.7 | 21.3 | 49.2 | 88.8 | 43.7 | 66.2 |
| | CNN (s) | Hyp-10-f | 73.2 | 70.8 | 10.8 | 14.7 | 18.9 | 23.0 | 96.3 | 88.1 | 74.5 | 72.7 |
| CIFAR10 | CNN (s) | Seed | 75.3 | 96.0 | 27.0 | 90.2 | 68.6 | 91.1 | n/a | n/a | n/a | n/a |
| | CNN (s) | Hyp-10-r | 50.1 | 88.0 | -4.3 | 40.5 | -2.7 | 34.2 | 34.0 | 50.5 | 71.5 | 80.9 |
| | CNN (s) | Hyp-10-f | 67.0 | 87.9 | 38.2 | 42.9 | 27.0 | 31.8 | 72.0 | 52.2 | 75.6 | 80.0 |
| CIFAR10 | CNN (l) | Seed | 83.6 | 98.2 | 33.4 | 92.9 | 86.5 | 95.7 | n/a | n/a | n/a | n/a |
| | CNN (l) | Hyp-10-r | 32.6 | 90.5 | -0.9 | 47 | -10.5 | 35.5 | 41.6 | 51.6 | 69.1 | 83.1 |
| | CNN (l) | Hyp-10-f | 64.5 | 91.4 | 30.4 | 40.7 | 29.8 | 35.3 | 74.5 | 54.9 | 77.7 | 86.0 |
| STL | CNN (s) | Seed | 17.8 | 91.2 | 2.0 | 30.2 | 45.3 | 95.0 | n/a | n/a | n/a | n/a |
| | CNN (s) | Hyp-10-r | -8.7 | 77.1 | -44.0 | 9.3 | -68.8 | 19.1 | 41.3 | 93.9 | 46.3 | 66.8 |
| | CNN (s) | Hyp-10-f | 76.1 | 76.5 | 6.7 | 10.7 | 21.2 | 22.4 | 98.1 | 91.3 | 78.1 | 62.6 |
| STL | CNN (l) | Seed | -112 | 94.2 | 2.8 | 37.3 | 5.6 | 98.7 | n/a | n/a | n/a | n/a |
| | CNN (l) | Hyp-10-r | -79.6 | 74.1 | -118 | 10.7 | -106 | 18.8 | 43.8 | 90.4 | 49.4 | 68.3 |
| | CNN (l) | Hyp-10-f | 84.1 | 77.7 | 10.4 | 11.7 | 14.6 | 19.1 | 97.8 | 92.8 | 78.8 | 68.0 |
| CIFAR10 | ResNet-18 | Seed | –.- | 96.8 | –.- | 76.7 | –.- | 99.6 | n/a | n/a | n/a | n/a |
| CIFAR100 | ResNet-18 | Seed | –.- | 97.4 | –.- | 95.4 | –.- | 99.9 | n/a | n/a | n/a | n/a |
| t-ImageNet | ResNet-18 | Seed | –.- | 96.1 | –.- | 87.5 | –.- | 99.9 | n/a | n/a | n/a | n/a |

Zoos with hyperparameter changes and random seeds are similarly unstructured. Zoos with varying hyperparameters and fixed seeds show clear clusters with models of the same initialization method and activation function. These findings are further supported by the predictability of initialization method and activation function (Table 3). The structures are unsurprising considering that the activation function is very influential in shaping the loss surface, while initialization method and the seed determine the starting point on it. Depending on the downstream task, this property can be desirable or should be avoided, which is why we provide both configurations.

**Model Property Prediction**   As a set of benchmark results on the proposed model zoos and to further evaluate the zoos, we use linear models to predict hyperparameters or performance values of the individual models. As features, we use the model weights $\mathbf{w}$ or per-layer quintiles of the weights $s(\mathbf{w})$ as in [55]. Linear models are used to evaluate the properties of the dataset and the quality of the features. We report these results in Table 3. The layer-wise weight statistics ($s(\mathbf{w})$) have generally higher predictive performance than the raw weights $\mathbf{w}$. In particular, $s(\mathbf{w})$ are not affected by using fixed or random seeds and thus generalize well to unseen seeds. For the ResNet-18 zoos, $\mathbf{w}$ becomes too large to be used as a feature and is therefore omitted. Across all zoos, the accuracy as well as the hyperparameters can be predicted very accurately. Generalization gap and epoch appear to be more difficult to predict. These findings hold for all zoos, regardless of the different architectures, model sizes, task complexity and performance range. $\mathbf{w}$ can be used to predict the initialization method and activation function to very high accuracy, if the seeds are fixed. The performance drops drastically if seeds are varied. This results confirms our expectation of diversity in weight space induced by fixing or varying seed. These results show i) that the model weights of our zoos contain rich information on their properties; ii) confirm the notions of diversity that were design goals for the zoos; and iii) leave room for improvements on the more difficult properties to predict, in particular the generalization gap.

# 5    Potential Use-Cases & Applications

While populations of NNs have been used in previous work, they still are relatively novel as a dataset. As use-cases for such datasets may not be obvious, this section presents potential use-cases and applications. For all use-cases, we collect related work that uses model populations. Here, the zoos may be used as data or to evaluate the methods. For some of the use-cases, the analysis above provides support. Lastly, we suggest ideas for future work which we hope can inspire the community to make use of the model zoos.

## 5.1    Model Analysis

The analysis of trained models is an important and difficult step in the machine learning pipeline. Commonly, models are applied on hold-out test sets, which may contain difficult cases with specific properties [31]. Other approaches identify subsections of input data that is relevant for a specific output [61, 25, 66]. A third group of methods compares the activations of models, e.g. the cka method used in Sec. 4 to measure diversity [27].

Populations of models have been used to identify commonalities in model weights, activations, or graph structure which are predictive for model properties. Some methods use the weights, weight-statistics or eigenvalues of the weight matrices as features to predict a model's accuracy or hyper-parameters [55, 12, 39]. Recently, [50] have learned self-supervised representation of the weights and demonstrate their usefulness for predicting model properties. Other publications use activations to approximate intermediate margins [58, 24] or graph connectivity features [6] to predict the generalization gap or test accuracy. Standardized, diverse model zoos may facilitate development of new methods, or be used as evaluation dataset for existing model analysis, interpretability or comparison method.

Previous work as well as the experiment results in Sec 4 indicate that even more complex model properties might be predicted from the weights. By studying populations of models, in-depth diagnostics of models, such as whether a model learned a specific bias, may be based on the weights or topology of models. Lastly, model properties as well of the weights may be used to derive a model 'identity' along the training trajectory, to allow for NN versioning.

## 5.2    Learning Dynamics

Analysing and utilizing the learning dynamics of models has been a useful practice. For example, early stopping [15], which determines when to end training at minimal generalization error based on a cross validation set and has become standard in machine learning practice.

More recently, methods have exploited zoos of models. Population based training [23] evaluates the performance of model candidates in a population, decides which of the candidates to pursue further and which to give up. HyperBand evaluates performance metrics for groups of models to optimize hyperparameters [34, 33]. Research in Neural Architecture Search was greatly simplified by the NASBench dataset family [59], which contains performance metrics for varying hyperparameter choices. Our model zoos extend these datasets by adding models including their weights at states throughout training, which may open new doors for new approaches.

The accuracy distribution of our model zoos become relatively broad if hyperparameters are varied (Figure 2). For early stopping or population based methods, identifying a good range of hyperparameters to try, and then identifying those candidates that will perform best towards the end of training, is a challenging and relevant task. Our model zoos may be used to develop and evaluate methods to that end. Beyond that, diverse model zoos offer the opportunity to make further steps of understanding and exploiting the learning dynamics of models, i.e., by studying the regularities of generalizing and overfitting models. The shape and curvature of training trajectories may contain rich information on the state of model training. Such information could be used to monitor model training, or adjust hyperparameters to achieve better results. The sparsified model zoos add several potential use-cases. They may be used to study the sparsification performance on a population level, study emerging patterns of populations of sparse models, or the relation of full models and their sparse counterparts.

### 5.3 Representation Learning

NN models have grown in recent years, and with them the dimensionality of their parameter space. Empirically, it is more effective to train large models to high performance and distill them in a second step, than to directly train the small models [21, 36]. This and other related problems raise interesting questions. What are useful regularities in NN weights? How can the weight space be navigated in a more efficient way?

Recent work has attempted to learn lower dimensional representations of the weights of NNs [18, 48, 63, 26, 50, 51, 52]. Such representations can reveal the latent structure of NN weights. Other approaches identify subspaces in the weight space which relate to high performance or generalization [56, 38, 1]. In [50], representations learned on model zoos achieve higher performance in predicting model properties than weights or weight statistics. [26] proposes a method to learn from a population of diverse neural architectures to generate weights for unseen architectures in a single forward pass.

Our model zoos can be either a dataset to train representations on as in [50] or [1], or as common dataset to validate such methods. Learned representations may bring better understanding of the weight space and thus help to reduce the computational cost and improve performance of NNs.

### 5.4 Generating New Models

In conventional machine learning, models are randomly initialized and then trained on data. As that procedure may require large amounts of data and computational resources, fine-tuning and transfer learning are more efficient training approaches that re-use already trained models for a different task or dataset [60, 14]. Other publications have extended the concept of transfer learning from a one-to-one setup to many-to-one setups [35, 53]. Both approaches attempt to combine learned knowledge from several source models into a single target model. Most recently, [51, 52] have generated unseen NN models with desireable properties from representations learned on model zoos. The generated models were able to outperform random initialization and pretraining in transfer-learning regimes. In [45], a transformer is trained on a population of models with diffusion to generate model weights.

All these approaches require suitable and diverse models to be available. Further, the exact properties of models suitable for generative use, transfer learning or ensembles are still in discussion [14]. Population based transfer learning methods such as zoo-tuning [53], knowledge flow [35] or model-zoo [47] have been demonstrated on populations with only few models. Populations for these methods ideally are as diverse as possible, so that they provide different features. Investigating the models in the proposed zoos may help identifying models which lend themselves for transfer learning or ensembling.

## 6 Conclusion

To enable the investigation of populations of neural network models, we release a novel dataset of model zoos with this work. These model zoos contain systematically generated and diverse populations of 50'360 neural network models comprised of 3'844'360 collective model states. The released model zoos come with a comprehensive analysis and initial benchmarks for multiple downstream tasks and invite further work in the direction of the following use cases: (i) model analysis, (ii) learning dynamics, (iii) representation learning and (iv) model generation.

## Acknowledgments

This work was partially funded by Google Research Scholar Award, the University of St.Gallen Basic Research Fund, and project PID2020-117142GB-I00 funded by MCIN/ AEI /10.13039/501100011033. We are thankful to Erik Vee for insightful discussions and Michael Mommert for editorial support.

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
