# OpenReview forum: "Model Zoos: A Dataset of Diverse Populations of Neural Network Models"
_NeurIPS.cc/2022/Track/Datasets_and_Benchmarks — NeurIPS 2022 Datasets and Benchmarks _

### Official Review · Reviewer_3DjW · 2022-06-30

**Rating:** 7
**Confidence:** 3

**Strengths:**

+ The Model Zoo datasets is a novel contribution to the analysis of machine learning model states and dynamics. It can also facilitate many real-world applications like Neural Architecture Search.
+ Model Zoo is constructed in a principled way, which well controls the individual impact of different datasets, model architectures and hyperparameters, and the model variance analysis is done under different diversity of random seeds.
+ The four proposed use cases are practical and interesting, which cover both the discriminative analysis of property prediction and representation learning and also cover the generative analysis of model design.


**Weaknesses:**

- The six selected image datasets and two selected CNN architectures are a bit out of date, which do not reflect the current trend in Computer Vision research with large-scale datasets like ImageNet and large-scale backbone model like Vision Transformer.
- In the supplementary material, authors state that Model Zoo is generated using CPU resources only, which neglects several important GPU training hyperparameters that play an important role in the model training of this era. For GPU-based training, models' states and dynamics could be influenced by mixed-precision training, activation checkpointing and how model states are synchronized across GPUs. These factors should also be taken into consideration when generating model zoos.


**Additional Feedback:**

In general, I admire the contribution of the proposed Model Zoo dataset. However, regarding the interests of modern machine learning model design, I have following questions for the author response period.

1. How will Model Zoo be extended to involve large-scale datasets like ImageNet and large-scale models like Vision Transformer? These involvements could attract more practitioners from the community.

2. How do authors plan to take the influence of GPU training hyperparameters into consideration?


**Clarity:**

The paper is well written, which makes it easy to capture the main ideas of this work.

**Correctness:**

The dataset is generated in a technically sound way with good controls on broad influential factors.

**Documentation:**

The official dataset and source code repository gives detailed guidance to reproduce the dataset generation process and utilizing the generated dataset.

**Ethics:**

I have no ethical concerns on this work.

**Relation To Prior Work:**

To my best knowledge, this dataset is a novel contribution to the related field.

**Summary And Contributions:**

This paper proposes a dataset of diverse populations of neural network models, called Model Zoo. In Model Zoo, six classical image datasets, two CNN model architectures and plentiful training hyperparameters are employed to generate the model checkpoints. These checkpoints record the whole training process of a specifically configured model on a specific dataset. Three different model zoos are constructed under the different diversity of hyperparameters and random seeds. The comprehensive model properties reflecting the performance and status of each checkpoint enable the downstream task of model property prediction, and the Model Zoo dataset can also be utilized for training dynamics analysis, model representation learning and model generation.

---

> ### Author Response · Authors · 2022-08-11
> **Response to Reviewer 3DjW**
>
> We thank reviewer 3DjW for the feedback, which we find encouraging.
>
> > W1: Limitations of Architecture and Dataset Complexity
>
> We have responded to  the concern regarding limitations in model and dataset complexity in the Response to all Reviewers under point 1.
>
> > W2: Impact of CPU / GPU Training
>
> That is a very interesting aspect of variation, which we have so far not considered. For the new, larger architectures, training on CPUs is unfortunately infeasible, but we will consider retraining one of the zoos with smaller architectures on GPUs. Further, we find the impact of full, half or automatic mixed precision on both model performance as well as structures in weight space an interesting mode of variation for future zoos.
>
> We look forward to the discussion, any further questions are very welcome!

---

### Official Review · Reviewer_hHSg · 2022-07-10
**A population of trained networks**

**Rating:** 6
**Confidence:** 4
**Correctness:** The paper is technically correct.
**Clarity:** The paper is clearly written and easy…

**Strengths:**

The populations of trained networks and their training trajectories on different datasets is a useful source of data for understanding the properties of the trained networks and their training trajectories.

**Weaknesses:**

(1) The properties such as diversities of the populations of the trained networks are more or less expected. It is unclear what deep insights can be gained from the proposed dataset. For a researcher interested in understanding the properties of the trained networks and the training dynamics, he or she may well design his or her own training protocols and collect his or her own dataset.

(2) The networks and the datasets are limited in their scales, which may limit the usefulness of the dataset.

**Additional Feedback:**

The authors may consider enlarging the scope of the datasets.

**Documentation:**

The documentation has sufficient details.

**Ethics:**

There are no ethical concerns.

**Relation To Prior Work:**

The related work is clearly discussed.

**Summary And Contributions:**

The dataset created in this paper consists of populations of convolutional networks trained on image classification datasets with different hyper-parameters, random seeds, and architectures. Various properties of the populations of trained networks are analyzed to illustrate the usefulness of the dataset.

---

> ### Author Response · Authors · 2022-08-11
> **Response to Reviewer hHSg**
>
> We would like to thank reviewer hHSg for the feedback and respond to the raised concerns below.
>
> > W1: Properties of Datasets / Significance
>
> The model zoos were designed to contain certain types of properties, in particular diversity. One can understand the lack of surprises as an achievement, as the zoos meet these requirements.
> We are of the opinion that the availability of such datasets can be beneficial for several use-cases as outlined in Sec. 4.. Of course, researchers may make the effort and train their own zoos. In computer vision, the availability of common datasets has greatly simplified research progress and lowered the entry barrier. Further, for comparability we feel it is preferable to have common, standardized zoos. We understand our submission and the blueprint of model zoo design, generation and analysis as a step toward that goal.
>
> > W2: Limited scale of model architecture and task
>
> We have addressed that concern in the Response to all Reviewers under point 1.
>
> Any further questions are welcome, we look forward to the discussion!

---

### Official Review · Reviewer_Kkhv · 2022-07-20

**Rating:** 7
**Confidence:** 4
**Correctness:** The claims made in this paper appear …
**Clarity:** The paper is very clearly written.

**Strengths:**

- Interesting, novel contribution with relevance to a broader research community.
- Large model zoo
- Clearly written paper.
- The data is easily accessible.

**Weaknesses:**

- The zoo only includes two very simple (e.g. in terms of parameter count) CNNs. In general, that is fine but the achieved performance is far behind SOTA models and implies that weights may not have been converged due to insufficient capacity of the model (see "Relation To Prior Work" [1]). This may have a negative impact of the dataset quality.
- The zoo is limited to low-resolution, classification tasks and only considers CNNs.
- Missing recommended documentations frameworks: datasheets for datasets, dataset nutrition labels, data statements for NLP, or accountability frameworks.
- The analysis is a bit short.

**Additional Feedback:**

Good paper.

**Documentation:**


- The dataset is sparingly documented, but sufficiently to be used. None of the recommended documentation frameworks were used.
- Long-term preservation is convincing.
- Licensed under CC BY-SA 4.0.

**Ethics:**

No ethical issues found.

**Relation To Prior Work:**

The paper clearly discussed previous contributions. I would suggest adding one more paper to related work that also performed an analysis on a large model zoo (albeit on a less systematically collected model zoo and limited to convolution weights):

[1] Paul Gavrikov, Janis Keuper, "CNN Filter DB: An Empirical Investigation of Trained Convolutional Filters", In Proceedings of the IEEE/CVF Conference on Computer Vision and Pattern Recognition (CVPR), 2022, pp. 19066-19076.

**Summary And Contributions:**

The authors provide a model zoo dataset consisting of  2.4M checkpoints of different architectures, trained on different datasets with different hyper parameters to provide a foundation for research into multiple areas including model analysis, discovery of unknown learning dynamics, representation learning, and generative modelling of neural network weights and biases. A small analysis is performed on the obtained dataset.

---

> ### Author Response · Authors · 2022-08-11
> **Response to Reviewer Kkhv**
>
> We thank reviewer Kkhv for the review. We appreciate the heartening feedback. We have addressed the concern regarding limitations in model and dataset complexity in the Response to all Reviewers under point 1 and will update related work. Any other questions are welcome, we look forward to the discussion!

---

> > ### Comment · Reviewer_Kkhv · 2022-08-14
> > **Long term preservation**
> >
> > Thank you for your response.
> > The NeurIPS guidelines mandate long term preservation: "Long-term preservation: It must be clear that the dataset will be available for a long time, either by uploading to a data repository or by explaining how the authors themselves will ensure this*.
> > I would argue that hosting (the new) data on Google Drive does is not long term preservation. Though, I fully understand that hosting datasets of this size is challenging. Would it be possible to split the dataset into many parts (e.g. on Zenodo) and then provide a download script that gathers them all?

---

> > > ### Author Response · Authors · 2022-08-15
> > > **Response #2 to Reviewer Kkhv on Long-Term Data Availability**
> > >
> > > Thanks for the response. We fully agree, the datasets must be available long-term and we are currently working to ensure that. The squeezed subset of the dataset (200GB) has been published on Zenodo, thanks to the help of the Zenodo team. It can be found at https://doi.org/10.5281/zenodo.6974028, the link has been added to the github repository.
> > >
> > > Further, we are working on a solution for the full CIFAR10-ResNet18 zoos. Regular Zenodo repositories are limited to 50GB, 200GB are an exception granted for a dataset linked to a publication. Splitting the zoo in 50GB chunks would require at least 46 individual Zenodo repositories, which we find impractical as well as against their fair use policy. However, we are currently in discussion with Zenodo to investigate alternatives and explore some of the available options we used for publishing the YFCC100m dataset [Tho16].
> > >
> > > [Tho16] YFCC100M: The new data in multimedia research, ACM Communication, 2016

---

### Official Review · Reviewer_KUo2 · 2022-07-25

**Rating:** 4
**Confidence:** 3
**Clarity:** See correctness.

**Strengths:**

- a collection of 47k+ unique neural model evaluations.
- empirical analysis of the performance across different model zoos
- provide an interesting use-case about learning model properties from the stored states.
- public repository for easy accessibility.

**Weaknesses:**

- The authors propose to generate the model zoos on 6 image dataset with three different configurations. The authors also suggest a triple of (dataset, hyperparameter, architecture), however the reported results in Table 1 suggest that only 2 of the 6 datasets have varying architectures. It is unclear how the architecture acts as a dimension in the model zoo.
- It is unclear how the results of Table 1 are obtained. Are they the average performance across different hyperparameters ? If so, it would be interesting to learn what the ranges of the hyperparameters are for HYP-10-Rand.
- Regarding HYP-10-FIX, authors state that they take the same grid as HYP-10-Rand, but re-run the same hyperparameter configuration, with a fixed seed 10 times. It is unclear what would play a role in the result discrepancy across multiple runs for the same seed/hyperparameter pairs.
- A key use-case of that could benefit from such a model zoo is hyperparameter optimization. It is unfortunate that the authors do no address it.


**Additional Feedback:**

See detailed comments.



**Correctness:**

The paper has alot of typos that makes it difficult to read.
- l7 : can be reveal
- l17 are generated and
- l26 these leads
- l46 disadvantage. (i), existing ..
- l155 a we document ..
- inconsistent use of hyperparameter and hyper-parameter


**Documentation:**

The documentation in the form of an open-sourced website.

**Ethics:**

I don't see any immediate ethical concerns.



**Relation To Prior Work:**

related.

**Summary And Contributions:**

This paper proposes a benchmark suite for the class of neural network models trained under different hyperparameter settings across a variety of datasets. This benefit of such a collection (model zoo) is to uncover topological structures in weight space, model analysis, and generative modelling. The authors also provide an analysis of the model zoo.

---

> ### Author Response · Authors · 2022-08-11
> **Response to Reviewer KUo2**
>
> We would like to thank reviewer KUo2 for the detailed feedback. We will revise the manuscript to remove typos and improve readability. We address the individual concerns below.
>
> > W1 Role of the Architecture
>
> We address concerns regarding limitations of the architectures and datasets in the Response to all Reviewers under point 1.
> In our submission, we use the triplet of architecture, hyperparameters and dataset to determine the components of model zoos. Within one zoo, we use only one architecture and dataset, to isolate the effects of hyperparameter changes. Several zoos can be combined to introduce other modes of variation, too. We have already extended the dataset with a zoo covering ResNet-18 architecture.
>
> > W2 Analysis Methods
>
> The analysis of models in the zoos is detailed in Section 3 “Analysis” in the submission. For the values in Table 1, all models in the zoos at their last epoch are considered. The metrics are computed to investigate the diversity of the zoos. The exact hyperparameter ranges for each zoo are detailed in Appendix A.
>
> > W3 Composition of Hyp-10-Fix zoos
>
> The composition of model zoos is explained in Section 2.1 “Model Zoo Design”, the aspect of random seed variation specifically in line 103. We will revise the submission to make this aspect even clearer.
> Both Hyp-10 repeat the same hyperparameter configuration ten times with different seeds. In Hyp-10-Rand, these seeds are randomly drawn for each configuration. In the entire zoo, models sharing the same seed are very unlikely. For Hyp-10-Fix, each configuration is repeated with the seeds [1,2,...,10]. That way, the zoos either incorporate effects of random seeds, or control for the exact seeds and thus isolate other variations.
> We will clarify this in the submission.
>
> > W4: Use-case Hyperparameter Optimization
>
> We agree with reviewer KUo2, hyperparameter optimization or neural architecture search are exciting potential use-cases for our model zoos. As they are, they can be used similarly to the NASBench family by ignoring the model checkpoints. Even more interesting might be the application for methods like population based training [Jader17], where models can be mixed or recombined in parameter space. Such methods may be refined with readily available, diverse models at different training stages.
>
> [Jader17] Population Based Training of Neural Networks. Jaderberg et al., 2017.
>
> We hope to have addressed the concerns, look forward to any other questions and the discussion.

---

### Official Review · Reviewer_Rxne · 2022-07-26
**A well documented dataset that needs a stronger justification of its significance**

**Rating:** 7
**Confidence:** 3

**Strengths:**

The details regarding the generation of the dataset are clearly presented so that researchers can feel confident using and reproducing the results. By defining a training trajectory using the triplet: dataset, hyperparamter, and architecture, the paper makes it easy to identify and reproduce specific training trajectories. Additionally, by presenting the statistics which measure the distribution of learning outcomes in terms of accuracy, pairwise weight similarities, etc., the paper clearly demonstrates the diversity of models contained in the zoo. The data is easily accessible online with transparent documentation to support its use.

**Weaknesses:**

Although this paper does a good job of clearly explaining and presenting the dataset, the significance of the contribution is at times unclear. For example, the introduction section hypothesizes that “NN models evolve on unique, smooth trajectories in weight space during training,” which “form topological structures in weight space,” and that this structure can “only be discovered” from the collection and analysis of model zoos. However, these claims are neither supported in the paper, nor are methods proposed for investigating these claims. Instead, section 3 focuses on characterizing the distribution of fully trained models from the zoo. This characterization supports the claim that the model zoo contains a diverse collection of trained models, but does not rigorously investigate the evolution of model weights during training or the geometric properties of the topological structures in weight space. While these analyses can and should be conducted in future work, the paper does not support the claim that these kinds of investigations can only be conducted using model zoos.

Section 2.1 states that “the model zoos have to be representative of real world models, but also diverse and span an interesting range of properties.” Despite this claim, the paper does not adequately justify the inclusion of the chosen datasets, hyperparameters, and architectures. Today, real world models tend to be very large with several orders of magnitude more parameters and hyperparamters than the CNNs training in this paper. Additionally, real world datasets in use today are often more complex containing many more examples than those used in this paper. If these models, hyperparameters, and datasets are not necessarily representative of real world models, then what commonalities do they share with those larger models? Why would we expect trends found in this model zoo to transfer to larger models trained on larger more complex datasets?


**Additional Feedback:**

NA

**Clarity:**

The paper is mostly well written, however, as stated previously, some claims are not fully supported in the text. Additionally, some grammatical errors exist and the paper would benefit from a thorough spelling and grammar check. For instance, “These leads” (line 26) should be “These lead.” “multiple training result” (line 29) should be “multiple training runs result.” “describes” (line 79) should be “described.” “To get a more in-depth insights” (line 183) should be either “To get more in-depth insights” or “To get a more in-depth insight.”

**Correctness:**

Yes, to the best of my knowledge.


**Documentation:**

Yes

**Relation To Prior Work:**

Yes, however it is unclear whether these differences constitute a significant contribution. For example, the introduction warns that existing model zoos suffer from “limited general utility” and that “a new model zoo causes subsequent work to lose comparability to existing research.” As stated in the weaknesses section above, it is unclear what characteristics help this dataset provide a general utility. Are these datasets, hyperparameters, and architectures the most generally useful to study? And, if not, what will make it possible to generalize the patterns discovered here to other datasets, hyperparameters, and architectures?

**Summary And Contributions:**

This paper presents a dataset, called a model zoo, of neural network training checkpoints and metrics. The networks are trained using a set of hyperparameters across a range of datasets. The authors hypothesize that the weights of populations of networks trained using the same hyperparameters form geometric structures in high dimensional space which can only be studied using model zoo datasets like the one presented here. The paper outlines the collection of 2,415,360 model states from training 47,360 neural networks on six datasets. Additionally, a preliminary analysis is presented to justify the collection of model zoo datasets.

---

> ### Author Response · Authors · 2022-08-11
> **Response to Reviewer Rxne**
>
> We thank reviewer Rxne for the review. We’re glad they find the dataset well constructed and presented. We address the concerns in the Response to all Reviewers and below.
>
> > W1: Significance of the contribution
>
> We have addressed that concern in the Response to all Reviewers under point 2.1.
>
> > W2: Model and Dataset complexity
>
> We respond to this concern in the Response to all Reviewers under point 1.
>
> > W3: Relation to Related Work
>
> There are many unstructured collections of NN models, e.g., model hubs, huggingface, or github repositories. However, to the best of our knowledge, there are only very few structured model zoos. Some of them contain only a few models [Jiang19], we are aware of only two larger collections of models. [Unter20] propose zoos of models of similar size as ours, but do not vary in the seed. [Eiler20] present zoos containing slightly larger models, but the dataset repository is not as accessible and well documented as it would need to be for reuse.
> The NASBench family of dataset is related to ours in that they are built by training large collections of models, however they contain pairs of architectures/hyperparameters and performance metrics, and not the actual model parameters.
> We chose the original model architecture for simplicity and parameter efficiency, and varied hyperparameters so that they would demonstrate a meaningful effect on the performance, but not (individually) break a model entirely. As we are scaling up models and datasets in complexity, we will rely on common architectures and attempt to represent hyperparameter (and architecture) variations that are used in the real world.
>
> [Jiang19] Predicting the Generalization Gap in Deep Networks with Margin Distributions. Jiang et al., ICLR 2019.

---

> > ### Comment · Reviewer_Rxne · 2022-09-02
> > **Improvements in claims and additional models**
> >
> > Thank you very much for your detailed personal and general responses. The addition of larger models to the zoo makes it more reflective of real world networks. In addition, the claims made in the introduction and future work sections now provide a more cohesive narrative regarding the contribution. I think that this has the potential to be a useful dataset in the areas outlined in the Use-Cases section. As a results, I will increase my score from a 5 to a 7.

---

### Official Review · Reviewer_ztuJ · 2022-07-26
**Review of "Model Zoo: A Dataset of Diverse Populations of Neural Network Models"**

**Rating:** 6
**Confidence:** 1

**Strengths:**

* This is a good initiative to provide a standardized model zoo dataset.
* I think this work may encourage future work on various related problems.
* The authors provided some analyses to investigate some initial results.

**Weaknesses:**

* There are some hypotheses/claims made in the paper that are not investigated or addressed in the end. For example "models evolve on unique, smooth trajectories in weight space during training", "would form topological structures in weight space", "We think that the geometry, curvature and smoothness of these structures contain information about the state of training and can be reveal latent properties of individual models", etc.
* The authors state that their dataset is likely to be useful in many tasks or use cases, however all of them are not backed by experiments. For example, reliability, bias, fairness, or adversarial vulnerability, etc., Section 4.3 (representation learning) and Section 4.4 (Generating New Models) demonstrated the utility of the proposed model zoos on none of the proposed tasks. Without experiments, it might be difficult to judge the usefulness. If a use case is not showcased in the paper, then it cannot be counted as a valid use case. Either the authors remove the unproven use cases into the "future work" discussion, or they provide more experiments to show the usefulness on these tasks.
* As training and saving many model checkpoints is not a difficult task in itself, it would be crucial that the model generating design is representative and diversified enough and the scale is large enough to convince other researchers to use this dataset as a standardized infrastructure. I acknowledge the efforts the authors made to promote the diversity in the model zoos, however I doubt whether it is enough. The 2 model architectures are tiny-sized (the larger one has about 10k parameters) compared to the actual models used in academia and industry. Each model is trained only for 50 epochs. The involved datasets consist of 6 image datasets with rather small-sized images. These factors harm completeness and diversity. The number of model checkpoints is large (around 2 million), but this number does not mean a lot since it is a matter of how much time to spend to generate model checkpoints. If the generating design is limited, it is like densely exploring a narrow search space but lacking the exploration elsewhere. Despite these problems, this work seems to be a good initiative.






**Additional Feedback:**

* Line 140, “The zoos include a total of around 47’360 unique neural networks.” Line 146, “approx. 2’415’360 collected model states.” The authors may want to provide the exact statistics instead of using words like "around, approx." Precise information will be more useful for user reference.
* Line 169: “A Tensorflow counterpart will follow.” The authors may want to promise an explicit time line for this follow-up work.
* Line 175: "Zoo cards with key values and visualizations are provided along with the zoos online." Please provide the link or precise access information as it is unclear where such information can be found.
* Line 178: "The threshold value is set for each zoo, such that it only excludes diverging models." The authors are invited to provide more information about how this threshold is set. This information should be useful to keep consciousness of potential bias when future works use this resource.


Typo:

Line 5: refereed => referred

Line 7: can be reveal => can reveal

Line 24 : high => highly

Line 26 : These => This

Figure 1: make available with all meta-data => made available with all meta-data

Line 79: describes => described

Line 119: Appendix C => Appendix A

Line 226: This => These

Line 256: then => than; raises => raise

Line 295: the => they



**Clarity:**

The paper is easy to read overall. However some writing could be improved, for example the paragraph from Line 147 to Line 150 is confusing.

**Correctness:**

See Weaknesses





**Documentation:**

The documentation is good overall. However it could be further improved, e.g. how to load the checkpoints, which checkpoints to choose if one has a specific task at hand.





**Ethics:**

Line 72, the authors stated “provide a ground for use beyond the defined benchmark tasks”, the authors may need to discuss potential misuse and measures to fight against them if relevant.

**Relation To Prior Work:**

The authors discussed many examples of model zoos in literature, but the comparison could be made more clear. For example, why is this new model zoo more general-purpose than previous ones, what are the drawbacks of previous work that could be fixed by this new model zoo dataset.


**Summary And Contributions:**

This paper proposed a dataset (referred to as model zoos) of model checkpoints, where each data point is a neural network checkpoint flattened into a real-valued vector. By setting up a generating design (hyperparameters, etc.), the authors generate a grid of neural networks and save the checkpoints at the end of each epoch (50 epochs in total). The authors believe that such zoos can help advance research in model analysis, investigating learning dynamics, learning rich representations and generative modeling of model parameters. The authors hope to provide a standardized dataset for these tasks. The proposed model zoo dataset involves 6 image datasets and 2 ConvNet architectures. An analysis is provided to explore the initial results of this dataset.


==================================================================================================

UPDATE:

After the rebuttal session, I increased my rating from 5 to 6 (in order to acknowledge the improvements made by the authors during the rebuttal).

---

> ### Author Response · Authors · 2022-08-11
> **Response to Reviewer ztuJ**
>
> We thank reviewer ztuJ for the feedback, we are glad they appreciate the initiative and potential such datasets hold. We will fix the typos in the manuscript and clarify potentially confusing sections. We have responded to the concerns in the Response to all Reviewers and below.
>
> > W1: Hypothesis and Claims
>
> We have responded to the concern in the Response to all Reviewers under point 2.1.
>
> > W2: Use-cases
>
> That concern, too, we have addressed in the Response to all Reviewers under point 2.2
>
> > W3: Model Architectures
>
> We answer the concern of small model architectures in the Response to all Reviewers under point 1.
> Regarding the number of epochs, we tuned the hyperparameters so that models approximately converge in 50 epochs. For future datasets, the situation may change, but until now, it has empirically worked well. The ResNet18 zoo on CIFAR-10 has approximately converged after 35 epochs, but we decided to continue training to also include that training phase in the zoo.
>
> > Ethics
>
> We agree with the other reviewers in that our datasets, the proposed use-cases as well as other use cases do not offer ethical concerns beyond regular machine learning. All zoos (both existing and planned) are trained on common datasets with common architectures, with the aim of understanding current models better and improving training regimes in the future. With that, model zoos are equally benign or malign as any generic machine learning method, and do not pose any particular risk of malicious misuse.
>
> We hope to have addressed the raised points, are happy to answer further questions and look forward to the discussion.

---

> > ### Comment · Reviewer_ztuJ · 2022-08-16
> > **Comment on the point 2.2**
> >
> > I appreciate the authors' efforts to extend model zoos to include more realistic models (ResNet18 trained on CIFAR-10), this is great progress.
> >
> > On the other hand, I would still recommend the authors to be more rigorous about the claims about use cases, e.g. by **explicitly** treating presumptive use cases as future (potential) use cases in the text. By presumptive use cases, I mean all the use cases that are not backed up by any experiments in this paper (even if they seem to be promising). I personally believe this can help set up a sound standard for dataset publication and thus help the community in general. Without experiments, it is hard to judge the usefulness on the claimed use cases.

---

> > > ### Author Response · Authors · 2022-08-16
> > > **Response # 2 to Reviewer ztuJ**
> > >
> > > We would like to thank Reviewer ‘ztuJ’ for the response. We appreciate that the reviewer considers the addition of the zoo of ResNet models to the submission ‘great progress’.
> > >
> > > We understand the reviewer’s request of being more explicit in potential use-cases for the proposed dataset. We will revise the manuscript (Section 4) to be more rigorous and mark use-cases explicitly as supported by experiments in this submission, as supported by related work, or as promising future work.
> > >
> > > We hope we could address the concern, and look forward to further questions.

---

### Author Response · Authors · 2022-08-11
**Response to all Reviewers**

We would like to thank all six reviewers (ztuJ, Rxne, KUo2, Kkhv, hHSg,3DjW) for their feedback.
We are glad that all reviewers highlighted the strengths of our dataset and paper. The reviewers underline the contribution *“good initiative to provide a standardized model zoo dataset”* (ztuJ), *“interesting, novel contribution with relevance to a broader research community”* (Kkhv), *“work may encourage future work on various related problems”* (ztuJ), the applications *“an interesting use-case about learning model properties from the stored states”* (KUo2), *“useful source of data for understanding the properties of the trained networks and their training trajectories”* (hHSg), *“novel contribution to the analysis of machine learning model states and dynamics”* (3DjW), *“can also facilitate many real-world applications like Neural Architecture Search”* (3DjW), *“four proposed use cases are practical and interesting”* (3DjW). Further, they point out the size and construction of the zoos *“Large model zoo”* (Kkhv), *“clearly demonstrates the diversity of models contained in the zoo”* (Rxne), *“Model Zoo is constructed in a principled way (3DjW)”*, *“clearly presented so that researchers can feel confident using and reproducing the results (Rxne)”*, the accessibility and documentation of the dataset *“public repository for easy accessibility”* (KUo2), *“The data is easily accessible”* (Kkhv), *“The data is easily accessible online with transparent documentation to support its use”* (Rxne), *“Clearly written paper”* (Kkhv).

Below we first address two shared concerns of the reviewers. Then we respond to the reviewers’ specific questions and concerns under the corresponding reviews.

---

> ### Author Response · Authors · 2022-08-11
> **1. Shared concern by ztuJ, Rxne, Kkhv, hHSg and 3DjW: “Size and complexity of models, datasets and tasks”**
>
> We intend the model zoos to be a living dataset that is constantly updated and extended with new populations. We understand this submission also as a blueprint on how to generate, analyze and publish model zoos. For that reason, we provide code to reproduce, correct, adapt or extend the zoos and invite the community to join in the effort.
>
> We agree with the reviewers ztuJ, Rxne, Kkhv, hHSg and 3DjW zoos of larger models are an interesting object of study. As a step in that direction, we add a new model zoo to this submission. The new zoo contains 1000 standard ResNet18 trained on CIFAR-10. Each ResNet18 model is 11M parameters large, rendering the entire model zoo 2.3TB large. Given the file limitation of 50GB (and 200GB as exception) of zenodo.org, we have uploaded a “squeezed” subset of the zoo to zenodo.org and make the full model zoo available on Google Drive. The subset at zenodo.org contains for each model the epochs 1, 10, 50, whereas the full 2.3TB is made available as 10 individual files of each 100 models containing all epochs (each 230GB large) at Google Drive.
>
> The corresponding evaluations of this new zoo can be found in the tables below and confirm the high degree of diversity as with the previous zoos. This zoo uses the same model zoo blueprint, but due to the size of the models is a considerable extension of the submission towards modern architectures.
>
> Table 1: extension of Table 1 in the submission: zoo diversity.
> | Dataset  | Zoo Configuration           | Accuracy   | aggreement  | cka-similarity | weight distribution | l2(w)-distance | cos(w)-distance |
> |----------|-----------------------------|------------|-------------|----------------|---------------------|----------------|-----------------|
> | CIFAR-10 | Resnet-18 Seed (11M params) | 92.1 (0.2) | 93.4 (0.7)  | TBD            | -0.01  (1.7)        | 122.1 (3.9)    | 72.2 (2.3)      |
>
>
> Table 2: extension of Table 2 in the submission: prediction of model properties.
> | Dataset  | Zoo Configuration           | Accuracy | Epoch  | GGap |
> |----------|-----------------------------|----------|--------|------|
> | CIFAR-10 | Resnet-18 Seed (11M params) | 96.8     | 99.6   | 76.7 |
>
>
> With that, we follow a roadmap to generate further model zoos. Additionally - but still ongoing - ResNet18 zoos are also trained on CIFAR-100 and tiny-ImageNet and will be added to the model zoo dataset once training is finished.
>
> Our intention with this dataset is similar to research communities such as Neural Architecture Search (NAS), Meta-Learning or Continual Learning (CL), where initial work started small-scale [Zhmo22,Rame22]. For example, in the NAS community, the  [NASBench datasets](https://www.automl.org/nas-overview/nasbench/) focusses on small-scale CNN models trained on datasets like CIFAR-10 or Fashion MNIST [YING19].
>
> Initial work in the area of learning from populations of neural networks models is currently based on similar sized datasets and architectures:
> - [Unter20] learn deep models to predict test accuracy (zoos of similarly small models, only weights and some data available, not the full models)
> - [Eilert20] predict hyperparameters from chunks of model weights (model zoos accessible, but not well documented and hard to make use of)
> - [Schur21] demonstrate that regularities in weights of NNs exist, can be learned in a self-supervised fashion and are good predictors for model properties
> - [Schur22] builds on the representation learning method, by sampling in representation space to generate new weights with targeted properties.
> -  [Zhmo22] use a HyperNetwork to generate weights for small-scale CNNs in a meta-learning setup.
>
> We are therefore of the opinion that our setup is comparable and therefore define a valid contribution to the proposed research directions.
>
> [Ying19] NAS-Bench-101: Towards Reproducible Neural Architecture Search. Ying et al., PMLR 2019
> [Unter20] Predicting Neural Network Accuracy from Weights. Unterthiner et al., Arxiv 2020
> [Eilert20] Classifying the classifier: dissecting the weight space of neural networks. Eilertsen et al., ECAI 2020.
> [Schur21] Self-Supervised Representation Learning on Neural Network Weights for Model Characteristic Prediction, Schürholt et al., NeurIPS 2021.
> [Zhmo22] HyperTransformer: Model Generation for Supervised and Semi-Supervised Few-Shot Learning. Zhmoginov et al., ICML 2022.
> [Rame22] Model Zoo: A Growing "Brain" That Learns Continually. Ramesh and Chaudhari, ICLR 2022.
> [Schur22] Hyper-Representations for Pre-Training and Transfer Learning, Schürholt et al., ICML Pre-Training Workshop, 2022.

---

> > ### Author Response · Authors · 2022-08-11
> > **2. Shared concern of ztuJ and Rxne: “Significance of dataset, assumptions, applications”**
> >
> > ### 2.1 Significance and Assumptions on Neural Network Training Trajectories
> > In the submission, we motivate model zoos given the hypothesis that training trajectories may contain information on the state of a model. We will revise the introduction in the manuscript to give more foundation for the arguments. In the following, we elaborate on how we came to these properties.
> >
> > This hypothesis on model training trajectories is based on related work. Smooth trajectories are intuitive given iterative, gradient based update schemes and some sort of momentum.
> > The **loss surfaces** of NNs are more and more well behaved, as the number of parameters grows [Good15, Dauph15, Li18]. Under reasonable assumptions on learning rate and step length, iterative optimization on such surfaces yield relatively smooth trajectories. As the **step length** along a **trajectory** as well as the curvature are determined by the change of the loss as well as how aligned the subsequent updates are, it is reasonable to us to assume that such information can be recovered from the trajectory [Caz22]. On the **uniqueness** of model trajectories: intuitively it is that as models often achieve similar performance and make similar errors, they end up in similar minima in weight space. Empirically, that only happens if models share architecture, initialization method and random seed. In experiments [Schur20], it was demonstrated that models trained with random seed remain far apart throughout training.
> > [Schur21], showed that structures in the weights of populations of models exist, can be learned and used to predict model properties. [Schur22] demonstrates that such beneficial structures can be instantiated in new models, as initialization for fine-tuning or transfer learning. Other work identifies structures in the form of subspaces with beneficial properties [Lucas21, Worts21].
> > With the proposed model zoos we would like to provide a foundation for systematic exploration of such properties.
> >
> > [Dauph15] Identifying and attacking the saddle point problem in high-dimensional non-convex optimization. Dauphin et al., NeurIPS 2015
> > [Good15] Qualitatively characterizing neural network optimization problems, Goodfellow et al., ICLR 2015
> > [Li18] Visualizing the Loss Landscape of Neural Nets, Li et al., NeurIPS 2018
> > [Lucas21] Analyzing Monotonic Linear Interpolation in Neural Network Loss Landscapes, Lucas et al., PMLR 2021.
> > [Worts21] Learning Neural Network Subspaces, Wortsman et al., PMLR 2021.
> > [Schur20] An Investigation of the Weight Space to Monitor the Training Progress of Neural Networks, Schürholt and Borth, arxiv 2020.
> > [Schur21] Self-Supervised Representation Learning on Neural Network Weights for Model Characteristic Prediction, Schürholt et al., NeurIPS 2021.
> > [Schur22] Hyper-Representations for Pre-Training and Transfer Learning, Schürhol et al., ICML Pre-Training Workshop, 2022.
> > [Caz22] Dataset Distillation by Matching Training Trajectories, CVPR, 2022
> >
> > ### 2.2 Applications and Potential Use-Cases
> > Model zoos as populations of neural networks are a rather new domain, therefore use-cases for such datasets may not be obvious. We agree that such use-cases are future work but similarly to dataset publications such as [Tho16][Koh21][Yeh21], we do want to give examples for potential use-cases for model zoos as done in Section 4.
> >
> > [Tho16] YFCC100M: The new data in multimedia research, ACM Communication, 2016
> > [Koh21] Wilds: A Benchmark of in-the-Wild Distribution Shifts, PMLR, 2021
> > [Yeh21] SustainBench: Benchmarks for Monitoring the Sustainable Development Goals with Machine Learning, NeurIPS Dataset Track, 2021

---

### Author Response · Authors · 2022-08-25
**Manuscript Revision, ResNet-18-CIFAR100 and TinyImageNet Zoos, and Sparsified Model Zoo Twins**

## Updated Manuscript
To incorporate the reviewers feedback, we have updated the submission. The changes are as follows
- **Introduction:** We have updated the introduction to address the concern of reviewers “ztuJ” and “Rxne”. It is more explicit in assumptions on model trajectories and extended with necessary related work as in our response to shared concern 2.1.
- **Model Zoo Generation:** Sections 2 and 3 and the Appendix have been extended by the ResNet-18 zoos.
- **Use-Cases:** We have revised Section 4 “use-cases” to address the concern of reviewer “ztuJ”. Section 4 now emphasizes if use-cases are based on previous work with model populations, on our own experiments, or are future work.

## ResNet-18 CIFAR100 Model Zoo
We are happy to report the second ResNet-18 zoo, trained on CIFAR100 has been added to the submission. A squeezed version is uploaded to zenodo, the full version is currently being uploaded. A third ResNet-18 zoo is currently trained on Tiny Imagenet, the first 115 models have been uploaded to Zenodo and added to the repository.

## Sparsified Model Zoo Twins
As mentioned before, our vision for the dataset is to continuously extend model zoos over time allowing the community to investigate different aspects of model populations. To that end, we are extending the model zoos with **sparsified model zoo twins** serving as counterparts to existing zoos in the dataset. Using Variational Dropout (VD) [Molch17], we sparsify the trained models from existing model zoos. VD generates a sparsification trajectory for each model, along which we track the performance, degree of sparsity and the sparsified checkpoint. Sparsified model zoos add several potential use-cases. The zoos can be used to study the sparsification performance on a population level, study emerging patterns of populations of sparse models, or the relation of full models and their sparse counterparts. Code and the first sparse zoo (MNIST Sparsified CNN-s) is already added to the collection. After 25 sparsification epochs, the models achieve on average 80% sparsity at only 2% accuracy loss. Sparse versions of the SVHN and CIFAR10-CNN(l) zoos will be uploaded in the coming days.

[Molch17] Variational Dropout Sparsifies Deep Neural Networks, Molchanov et al., ICML 2017.


With that, we hope to have incorporated the feedback and improved the submission. If there are any remaining questions, we are happy to answer them and look forward to the remaining week of discussion.

---

### Meta-Review · Area_Chair_byub · 2022-09-08

**Recommendation:** Accept
**Confidence:** 5

**Metareview:**

This study constitutes a gargantuan empirical effort to characterize a large number of neural network models and parameters (referred to by the authors as the model zoo). The dataset includes 27 model architectures and a total of 50'360 models when considering all hyperparameters. This large empirical investigation can be useful as benchmark for multiple tasks and also to further understand the dynamics of training in neural networks. It can also serve the purpose of testing more theoretical approaches to study learning in neural networks.

---

### Decision · Program_Chairs · 2022-09-16

Accept